# Challenges of the Implementation of a Delirium Rate Scale in a Pediatric Intensive Care Unit: A Qualitative Approach

**DOI:** 10.3390/healthcare12010052

**Published:** 2023-12-26

**Authors:** Paula Balsalobre-Martínez, Raquel Montosa-García, Ana Marín-Yago, Manuel Baeza-Mirete, Gloria María Muñoz-Rubio, Andrés Rojo-Rojo

**Affiliations:** 1Hospital Universitario Virgen de la Arrixaca, Public Murcian Health System, 30120 Murcia, Spain; 2Pediatric Intensive Care Unit, Hospital Universitario Virgen de la Arrixaca, Public Murcian Healthcare System, 30120 Murcia, Spainana.marin@carm.es (A.M.-Y.); 3Faculty of Nursing, Catholic University of Murcia (UCAM), 30107 Murcia, Spain; 4Intensive Care Unit, Hospital Universitario Virgen de la Arrixaca, Public Murcian Health System, 30120 Murcia, Spain

**Keywords:** Pediatric Intensive Care Unit, Delirium, delirium rating scale, CAPD, pediatric nursing critical care, implementation program

## Abstract

Introduction: Delirium in the pediatric population admitted to intensive care is a worrying reality due to its potential complications and the increase in associated costs. This study aims to explore the experiences of nursing staff of a Pediatric Intensive Care Unit after 15 months of starting a program to fight against childhood delirium in their unit. Methodology: A qualitative study was conducted through semi-structured interviews with Pediatric Intensive Care Unit (PICU) Key Informants. The Standards for Reporting Qualitative Research (SRQR) and the consolidated criteria for Reporting Qualitative Research (COREQ) were followed as quality measures for the study. Seven nurses (33% of the eligible population) from the PICU of a referral hospital were interviewed. Text transcripts were analyzed using the Interpretative Description and Qualitative Content Analysis method. Results: The interviewees indicated not identifying delirium as an important reality; with great deficiencies observed in what is related to the identification of delirium; identifying CAPD as an unreliable tool in their unit; and not sharing therapeutic objectives in this respect with the medical staff. Conclusions: The nursing staff presented a series of negative attitudes towards the phenomena of delirium in their unit, with gaps in training and in clinical management, and the diagnostic tool used, and did not see it as a priority objective of the unit, partly due to a resistance to change and a latent interprofessional communication conflict. A change at the formative, attitudinal, and relational levels is urgently needed for the success of the program and the well-being of the children in the unit.

## 1. Introduction

Delirium is a neuropsychiatric syndrome with decompensation of brain function, characterized by acute cognitive changes, arousal states, motor agitation, dementia in-stability, bradypsychia, among other manifestations [1,2]. The DSM VI considers the diagnosis of “delirium” as an acute alteration of consciousness accompanied by inattention, disorganization of thought, and alterations of perception that fluctuate in a short period of time [3]. The ICD 10 considers delirium with the synonym of “Acute Confusional Syndrome”, as an acute, transitory, global and reversible condition that affects the consciousness of the subject who suffers from it [4].

In the ICU, delirium results from a serious underlying medical condition or side effects of treatment [5].

Delirium in a critically ill adult patient has been recognized as a problem, with a prevalence ranging from 25–87% [6,7,8]. This variability is explained by the different types of patients and pathologies, with age and hemodynamic deterioration as influential factors.

In the pediatric population, prevalence studies published to date show a range between 25% and 69% [9,10,11] of admitted patients. The use of sedative drugs [12] and opioid analgesics has been identified as one of the predisposing factors to the occurrence of delirium in a critically ill pediatric patient as a consequence of the different procedures performed in the unit.

Delirium in both adults and children is associated with a prolonged hospital stay, increased morbidity and mortality, increased workloads, and consequent increased social and health care costs [13], and decreased quality of life for the patient and/or family [14,15,16,17,18].

Among the guidelines provided in 2013 for the management of the impact of delirium in critical patient units, by the Society of Critical Care Medicine [19], an affiliate of the American College of Critical Care Medicine, we find the use of validated tools for the measurement of delirium and pain for its identification, and therefore, for actions in initial phases and identification of predisposing factors.

Among the tools validated for the assessment of delirium in critical pediatric patients, we find the “Cornell Assessment of Pediatric Delirium” (CAPD) [20], the “Pediatric Confusion Assessment Method for the Intensive Care Unit” (pCAM-ICU) [21], and the “Preschool Confusion Assessment Method for the Intensive Care Unit” (psCAM-ICU) [22]. The latter are adaptations for the pediatric population of the Assessment method from the “Confusion Assessment Method for the Intensive Care Unit” (CAM-ICU) [23], originally created for the identification of delirium of critically ill adult patients. All of these versions are presented in their original versions, as well as versions adapted to other languages, such as Spanish [24,25,26], which eases their use and their independent application within the cultural context of the present study.

Nevertheless, the use of tools such as the CAM-ICU or CAPD by health professionals requires a strong orientation towards the identification of the problem, which implies a change of attitude towards it, and the applicability of the tool: how it will be utilized, the definition of the process of identification and recording of the basal mental state, and the definition of how the documentation will be crafted [27].

Despite this significance, health professionals [28] feel that delirium is a complex situation of a patient who needs care, but do not perceive it as an urgent condition that must be resolved immediately, which results in variable management strategies.

Intensive care nurses are recognized as nurses with skills and opportunity to apply these tools, given their presence at the bedside of the patient, and the possibility of recording the changes in behavior of patients with delirium [29,30]. Likewise, the fact that the diagnostic instruments be validated and possess excellent psychometric qualities make their use possible by any health professional [31].

Considering the above, and the recommendations from the Society of Critical Care Medicine and the Spanish Society of Pediatrics, in January 2020, a program of early detection and care of delirium was implemented within the Pediatrics ICU at our hospital. At the start of the program, no previous data was available with respect to the prevalence of delirium in our unit, given that the use of the detection tools was not systematic with all the children admitted.

The program was designed considering the pediatricians belonging to the unit. It was based on the systematic application of the delirium scale CAPD to all the children admitted to the unit, independently of their pathology or treatment. The application was performed by the nurse caring for the child, at admittance into the unit, and posteriorly in a routine manner every 12 h and/or when a change in the child’s behavior was observed. The scale had to be administered by the nurse responsible for the care of the child during working hours. Thus, it was listed as an additional task in the routine care of the child. This tool was integrated into the Phillips-IntelliSpace Critical Care and Anesthesia^®^ (ICCA^®^) program, for its computerized recording by the nurse.

Data analysis at 6 and 12 months after application was established as control and follow-up points. The implementation of the protocol was not agreed-upon with the nursing staff, and no specific training was provided on the use of the selected tool. Nevertheless, all the health professionals were aware of the existence of the tool.

The emergence of the COVID-19 pandemic resulted in the lack of consideration of the control points expected for June of 2021. In January 2022, a data analysis was performed, which indicated the presence of delirium in 56% of the children admitted. Nevertheless, it was only implemented in 45% of the patients admitted, so that the data were not concluding, with respect to the prevalence of delirium in the unit. At the time of the study, the completion of the CAPD tool is still an obligatory requisite as part of the care of children admitted to the unit.

Given the data collected, and the worrying low completion of the scale, the objective of the study was to identify the attitudes, values, and beliefs of the pediatric ICU nursing personnel towards the program of detection of acute delirium in treated children, and the use of the delirium diagnostic tool.

## 2. Materials and Methods

### 2.1. Study Design

The research is based on semi-structured interviews with key Informants in the intensive care unit. The analysis of this qualitative study is based on an Interpretive Description (ID) [32] to understand the work of nurses in the Pediatric Intensive Care Unit (PICU). ID research describes and interprets a phenomenon, considers the meaning of related behaviors, and admits reality as socially constructed, and that reality involves multiple constructed realities, while also recognizing that experimenter and party influence one another. ID studies provide a thematic description of a phenomenon of interest [33], in this case, to identify the impressions and experience related by the nursing staff about the phenomenon of delirium in pediatric patients attended in the intensive care unit.

ID, which is widely used in the qualitative exploration system within nursing, offers an accessible and theoretically flexible approach to analyzing qualitative data within the exploration of the phenomena studied. ID is a methodological approach applicable to the exploration of healthcare professional education, as it can address complex existential questions and at the same time raise practical questions.

The performance and reporting of this study were guided by the Standards for Reporting Qualitative Research (SRQR) [34] (See it at Appendix A). In addition, the authors followed the guidance offered by the Consolidated Criteria for Reporting Qualitative (COREQ) research checklist (See it at Appendix A) [35].

### 2.2. Setting and Context

Our hospital has a multi-purpose, level III Pediatric Intensive Care Unit, able to tend to 13 children aged between 0 and 18 (according to the pathology). This ICU is considered first-class for the treatment of children in the Region of Murcia (Spain), which has a population of 1.5 million people. The rate of bed occupation is 87% during the year, and it employs 25 registered nurses (RN) and 4 pediatricians specialized in pediatric intensive care. All of the nurses who work in this unit are specialized in children’s nursing, as stated in the current legislation on health professionals in Spain [36]. The unit has a Phillips-IntelliSpace Critical Care and Anesthesia^®^ (ICCA) computerized system for recording the clinical history and medical and nursing treatment of admitted patients, which speeds up the documentation and communication between the different health professionals.

### 2.3. Sample

The sample consisted of nursing staff belonging to the PICU of a first-class hospital in southeastern Spain. Participants were recruited by convenience sampling through the Snowball sampling method. Snowball sampling can be used effectively to analyze vulnerable groups or sensitive or socially questionable situations [37]. Researchers often start with a small number of initial contacts that fit the research criteria and use the social contacts from this initial set to increase the number of research subjects [38].

We understand a “Key Informant” (KI) as a person who possesses exceptional knowledge of a situation, given their personal skills or position within a society, and can provide further information and insight of what is occurring around them [39].

As inclusion criteria, we considered the participants who work as registered nurses in the unit for at least three years, as they are considered KI. No exclusion criteria were defined. At the end of the interview, each KI provided two names of persons belonging to the unit who met the inclusion criteria, thus complying with the snowball sampling system.

The study was carried out in a first-class PICU for a population of approximately 1.5 million inhabitants, with 13 multi-purpose beds, where children with critical pathologies from 21 days of life to 14 years of age are cared for. The nursing staff comprised a total of 25 nurses specialized in pediatrics, together with 10 auxiliary nursing care technicians (healthcare assistants), and 8 pediatricians specialized in critical care. More than 80% of the nurses working in this unit have the title of “Nurse Specialist in Pediatrics”, as recognized by the Spanish Ministry of Health. More than 80% (N = 21) of the nurses in this unit have more than 3 years of continuous experience in the unit.

In qualitative research, the number of subjects is adjusted according to data saturation [40,41]; data saturation is the term used to describe the phase of data collection in which no further information emerges from subsequent collection efforts, meaning that continuing to explore more themes does not provide modification of the data already obtained.

The current evidence has not reached a consensus on the minimum number of subjects to reach saturation, with reports indicating that 4–6 key informants are sufficient [42], while others increase this number up to 12 [43].

After the data collection process, seven nurses were selected as research subjects. They represented approximately 33% of the total PICU nurses analyzed.

### 2.4. Data Collection

A qualitative, semi-structured interview was conducted (Appendix A). The interviews focused on how the nurses understood delirium in child patients admitted to their unit, the use of any tools for its termination, their degree of acceptance of its use, and difficulties inherent in its use.

The interview script [44] was created based on the purpose of the research and the literature review, and included the following four main axes summarized in the following topics: (1) What do you think about the phenomenon of delirium in children in your unit? (2) Do you believe that its incidence is higher in recent times; Why? (3) As a nurse, do you have any kind of tool to combat this phenomenon? (4) Do you feel that working against delirium is a multi-professional objective in your unit? Do you feel involved as a professional or is it something outside your work as a nurse? (5) Do you have sufficient training and skills to care for children with delirium? Do you feel confident in your professional skills to identify and handle a child with delirium? (6) Concerning the experience you have had following the care of a child with delirium, do you have any other thoughts you would like to share with me? This interview script was not tested previously. It was only checked by the senior researcher of the research group.

Before interview, the research team (ARR & PBM) contacted the proposed research subjects and arranged a suitable date and time for the interview, after explaining the procedure to follow and the objectives of the study to them. Contact was conducted face to face.

The interviews were performed during the professionals’ working day, using time slots with low work-loads in an environment with adequate privacy and sound conditions. Their information was collected in a location inside the PICU itself, where private information is usually exchanged between the parents of the children admitted and the healthcare staff, using respiratory protection measures (face mask), as required by the post-COVID pandemic situation and the clinical environment where the interview was carried out. The minimum distance measures were maintained, and physical contact was avoided between the interviewer and the interviewee.

The nurses were interviewed by the first author. The interview was carried out only once for each interviewee, without interruptions. The interviewer was present at all times, no other person was present during the interview, and there were no repeat interviews. The session began with a welcome to the participant, followed by the reading and signing of the informed consent form. The interview began with initial questions about the participants’ work, followed by the central themes. The participant could stop the interview at any time if they were uncomfortable. Participants were informed of the option to withdraw their testimony from the study at any time until their data were included in the analysis phase. No selected nurse renounced participation.

All the interviews were conducted throughout the month of June 2021 (1–30 June 2021) and lasted between 45 and 150 min.

The interviews were audio-recorded and transcribed by the research team. The investigator took field notes to support the discourse of the interviewer. The transcripts were supplemented by the field notes taken by the interviewer. The interviewees were coded with the acronym NUR followed by the interview number temporarily. In this way, the subject interviewed first was coded with the code “NUR1”, the second with “NUR2”, and so on. This coding allows the data to be analyzed on the basis of the interview order, so that the timing of data saturation could be precisely adjusted. The identity of the interviewees was not transferred to the transcripts texts; only age, gender, length of time working as a nurse, and length of time spent in the PICU were collected as personal variables, with the intention of providing qualitative aspects of the sample analyzed.

Neither the transcripts nor the codes were reviewed by the interviewees.

### 2.5. Data Analysis

Four researchers were involved throughout the collection and analysis process to ensure the reliability and objectivity of the process. Two of them (RMG & AMY) have a Master’s degree in nursing, as well as a specialization in pediatric nursing, and experience in the unit where the study was carried out. The third (PBM) was in the process of obtaining a Master’s degree in nursing. Finally, the process was supervised by a researcher (ARR) with first-hand knowledge of the situation in the study unit, experience in qualitative research methodologies, and a PhD degree.

The analysis was carried out using a Qualitative Content Analysis Method (QCA) of the transcribed discourse [45]. The QCA method is defined as “the systematic reduction of content, analyzed with special attention to the context in which it was created, to identify themes and extract meaningful interpretations of the data [46] (p. 361)”.

The two-phase, eight-step process associated with conducting a QCA [44] was followed; in phase 1 (data generation), the same researcher (PBM) who conducted the interviews listened to the interviews several times, transcribed and typed them, including interpretative aspects such as gestures, emotional aspects of the interviewee, and the researcher’s perceptions during the interview. In phase 2 (data analysis), another two researchers (RMG & AMY) monitored the categories found in phase 1, analyzed the relationships between the different categories and established the contextual interpretations and implications of the testimonies collected. The entire process was supervised by the researcher (ARR) with experience in qualitative methods.

Discrepancies that arose in the process of coding and/or interpreting the codes were resolved through a group discussion of the four researchers responsible for the analysis. The research questions were taken as initial themes or categories and formulated as preliminary hypotheses. The definitions of the initial categories and themes were discussed with all authors to arrive at a final version.

No software was used for the technical coding and classification of the data.

### 2.6. Trust and Reliability Strategist

We ensured the rigor of the data analysis with the three researchers working independently in the initial phases of the process; the data were extracted and coded (phase 1) by a researcher who was not involved in the conceptual phase, avoiding possible contamination of the process. Phase 2 was carried out by two researchers in parallel and independently, who were familiar with the reality of the situation under study. After the completion of phase 2, the results of the analyses were discussed as a group by the three researchers, together with the senior researcher, identifying overlapping interpretations and reaching consensus on diverging interpretations [47].

We understand that this process ensures that the data analysis was not contaminated by qualitative research biases such as observational bias. In addition, the use of the SRQR and COREQ guidelines ensured that the research process was validated.

### 2.7. Ethics Approval and Consent to Participate

Ethical approval for this study was obtained through the Ethics Committee of the Catholic University of Murcia (UCAM), protocol report number CE06211. The participants gave their informed consent in person by signing a consent form and through verbal consent recorded at the beginning of the interview. No participants withdrew from the study. To maintain participant confidentiality, the participants were coded in order of interview, with an alpha-numeric code identifying the category and order.

## 3. Results

Seven nurses were interviewed in the selected study period (see Table 1 for characteristics). Their average age was 32.5 years (25–42), with an average experience of 11 years (4–23), and an average experience in the PICU of 7 years (1–15). All the participants were female; all of them were pediatric nurse specialists; but 70% of them obtained the specialization of pediatric nursing through the specific training program of Resident Internal Nurse, according to the current Spanish legislation. Those who did not take part in this training program obtained this recognition given their experience in the unit.

A total of 15 code categories were identified, grouped into four main themes; some categories were emergent while others were raised in the interview script. (Table 2 for an overview). The themes will be described below, using quotes to illustrate the interviewees’ experiences.

### 3.1. Childhood Delirium in the PICU

This theme describes the experiences and impressions of the participants about the central theme of the study. This theme captures an important core of issues, which raised a priori. The general impression of the participants is that the number of children diagnosed with “delirium” has been increasing over time, especially in the last year (2021).


*“…it is fashionable; there is no child who has been with us for a while, for whatever reason, who is not labeled as having delirium”.*
NUR 6


*“…it seems to be fashionable”.*
NUR 2


*“…I have never seen as many children diagnosed with delirium,…as I have seen so far”.*
NUR 7


*“I hadn’t heard the word delirium for at least a year and a half or two years,…… I heard kids with “withdrawal”, taking methadone and stuff. I don’t think I saw any hypoactive delirium either. They were all clear withdrawal syndromes. So I think there’s been some change in the way we diagnose or something,…. Or what used to be called Withdrawal Syndrome is now called delirium.*
NUR 3

The participants understand this fact as a diagnostic desire from the doctors rather than the real existence of the symptoms. They understand that it is a disproportionate interest that overestimates the presence of delirium, when what the child truly has is a process of dehabituation to morphine-derived drugs. It is therefore clear that nursing professionals are not involved in the diagnosis of this condition.


*“…it is not possible that depending on the doctor on duty, children are classified as delirious, and if there is another doctor, they are not. Perhaps it is because one has a perception that the other does not, and that is why there are more deliriums now than before….. when the children are in the same situation”.*
NUR 6


*“Nursing professionals must be taken into account because we are with the child 24 h a day,… but we cannot establish a diagnosis; that is the doctor’s job”.*
NUR 3


*“we can tell the doctor what we think about the diagnosis of delirium in a child… but it is up to him (or her) to label it or not”.*
NUR 2

Regarding the identification of the patient with delirium, some testimonies identify situations compatible with delirium,


*“For me, … a child with delirium, …well, it is a child who has been sedated, who has just woken up and is conscious but is not connected. He is therefore looking up a lot. He doesn’t communicate well and that’s why he cries and is nervous…”.*
NUR 6

In terms of treatment, the use of pharmacotherapy is the main pillar on which the interviewees state that the unit’s doctors treat delirium. Comfort or non-pharmacological measures are left to the initiative of the nurses working at the time and are not related to the identification or not of the child as a patient with delirium, but rather to general or preventive measures.


*“Delirium is treated with medication prescribed by the doctor. Normally with levomepromazine”.*
NUR 1


*“I think they put him on Sinogan. Basically. I don’t know if they put him on something else. I think… dexmedetomidine”.*
NUR 5


*“Well, we try to make them feel as comfortable as possible. But I don’t distinguish between those who have delirium and those who don’t. I think it’s good for all children, a quiet environment, with little noise, and the parents present with them. It calms them down”.*
NUR 3


*“If we can do something… And we have time. So… we do those things too”.*
NUR 6

### 3.2. Childhood Abstinence Syndrome in the PICU

This theme emerges as a category in the discourse of several interviewees. One is seen as an error in the identification of one diagnostic label with another. The nursing professionals understand that most of the children diagnosed with delirium are suffering from withdrawal syndrome derived from the use of morphine-derived medications in children in hospitals.


*“Perhaps we lack more information or more understanding of the difference between delirium and withdrawal syndrome. I don’t see it very clearly nowadays”.*
NUR 5


*“There is a bit of confusion as to what is or isn’t delirium. That is confused with what is withdrawal syndrome”.*
NUR 2


*“All children who used to have withdrawal syndrome and now what is called delirium. Normally hypoactive, which is true that we didn’t see it before,…… Of course, that’s what it is for me, in the other (hyperactive delirium) it would be difficult to differentiate it from withdrawal syndrome. For me. I haven’t received any training”.*
NUR 1


*“And from a year ago, a lot of children have been seen… …Yes, there is a difference between withdrawal syndrome and hyperactive syndrome. Because I do notice that some tremors and things are different from withdrawal, and the other way around. The hypoactive one, you can see that the child is… that he doesn’t move, he’s looking at the moon… And you can see that it’s something”.*
NUR 3


*“The concepts are not clear. In fact, not even for them. Because of the doctors, there are eight who still talk about withdrawal syndrome and there are two who are in charge of delirium and are the ones who diagnose it”.*
NUR 6

### 3.3. Use of Diagnostic Tools

Another issue to be highlighted in the interviews is the use of scales for the detection of delirium in the pediatric patient. Of the multitude of scales that have been validated for the detection of delirium in the pediatric population [31], all the nurses interviewed indicated that the specific scale for delirium is the Cornell Assessment of Paediatric Delirium (CAP-D). Although this scale is adapted to the specific nursing role, all interviewed nurses state that the tool has a high inter-observer variability, which depends on the subjectivity of the observer, and the characteristics of the individual patient. In addition to this criticism, based on their own experience, the nurses also point out that there is a lack of knowledge among the nursing staff about the use and handling of the scale in particular, and about delirium in general.


*“I have to apply the delirium scale to a girl who is intubated, sedated and relaxed. So I’m not going to be able to answer most of the things. What’s going to come out of it? It’s not assessable.”*
NUR 1


*“Sometimes the parameters that are asked are not appropriate for all age groups”.*
NUR 3


*“Well, … I don’t see them (delirim scale) as an objective tool at all. I don’t know how to assess … I don’t know what score to give the child, …?”*
NUR 6

*“The delirium scales you have to make them up because you can’t complete them if you don’t make them up. Because they are not real. That’s what I think.”*.NUR 5


*“For me that tool is useless. It’s that, really, it’s also very … never, rarely, such … I mean, I think that’s what I think. Maybe their mother, who is at the bedside, can give you an answer, and she will also be wrong”.*
NUR 4

The interviewees state that there are no criteria to use the scale; in their clinical setting, it is indicated by the pediatricians that the delirium detection scale is applied for all patients without distinction. However, they add that nursing should take part in the decision on whether to implement the scale, as there are occasions when it would not be necessary due to the absence of risk factors in the patient. They feel left out of such decisions and are merely executors of doctors’ orders.


*“While the study was being carried out, it had to be done systematically to all children, even if they had a catheterisation, and they didn’t need so…beasuse the child was aware…”.*
NUR 5


*“It goes like a packet. Just as if they (the doctors) order us to put a nasogastric tube into the child, they order us to assess the delirium scale. But it is not necesary … We know when the delirium scale should be applied to a child, and when it is not needed …. We know what a valid measurement is, and when it is not”.*
NUR 1


*“We have the right to decide that this child doesn’t have to pass the delirium scale, because he doesn’t have delirium. He is not taxed and others are”.*
NUR 2

### 3.4. Professional Training

The training of nursing staff in delirium is a key aspect for providing quality care. The team that was interviewed indicated that they had little training on the subject and that the information received from the unit was scarce.


*So they (…the doctors…) have started to train themselves, they introduce the scales to us and from there they get into training that we…they don’t take into account.*
NUR 3


*Maybe we have little training on delirium. Nowadays, this subject is more in vogue, and I do think it is important to deal with it. But I think we need a bit more… We need to know what delirium is and how it is treated.*
NUR 1


*Nothing has been explained to us. We have a video there, which they play it for us to watch, but apparently it couldn’t be opened. I asked recently and he told me that yes, I could watch it, but in the end we haven’t tried it. But it wasn’t training either, it was just a video. Here. You know?*
NUR 3

## 4. Discussion

The present study helps us to better understand the experiences of nurses working in the PICU regarding the phenomenon of delirium in the critically ill pediatric population cared for in these units. The multidimensional nature of this pathology makes nursing work crucial in both management and diagnosis [48].

If we consider the low quality of the data obtained in relation to the low degree of completion of the CAPD records, they are not surprising. The fact that the COVID-19 pandemic burst into the area of healthcare at that time could have affected the implementation of detection measures against delirium, as it diverted the attention of all healthcare providers to other, higher-priority areas. However, the incidence of COVID patients in the unit was lower than in the adult units. The National Epidemiological Surveillance Network of Spain [49] indicated that only 0.6% of pediatric patients affected by COVID (positive in a detection test and with symptoms of the illness) required hospital admission, and of these, only 5.2% were admitted to the ICU, with these figures being much lower in relation to those obtained in the adult population. For this reason, we consider that the impact of COVID-19 does not justify not completing the detection records; on the contrary, if it had been carried out, it could have contributed to linking the presence of delirium in the child with COVID-infected children admitted to the ICU.

Some authors point out that despite international recommendations on the impact and measures to be considered to prevent delirium in critically ill patients, both in adult and pediatric patients, the use of these measures, especially the use of validated diagnostic tools, is very limited. Patel [50] conducted a survey of 1384 professionals from various ICUs in North America and revealed that more than half of them (59%) assessed their patients for delirium, although only 20% used a valid evaluation tool. Luetz [51] investigated the implementation rate of delirium monitoring in several hospitals through anonymous surveys and questionnaires, with similar results (a 27% implementation rate of delirium assessment with a validated tool).

The recorded testimonies indicated that the nurses in this PICU identified that behavioral problems similar to delirium already existed before starting this program, which was nothing new. They indicate that the label “Delirium” is not entirely appropriate, and that the phenomenon is exaggerated, not agreeing with the use of this diagnostic label; that the indiscriminate use of delirium detection tools is “absurd” and a “waste of time”. For most of the interviewees, the diagnostic label of delirium was, in fashion, associated with medical approaches, and not a reality that they had to tackle. Delirium was not highlighted as a concern that nursing must address imminently.

In this sense, their opinions towards the phenomenon of delirium and the use of identification scales coincide with previous studies [52,53] that analyzed the application by intensive care nurses of pain, agitation, and delirium scales, finding that only 41% felt comfortable or very comfortable with the use of delirium measurement tools, as compared to the use of pain and agitation scales that showed satisfaction figures greater than 80%. Likewise, they indicate that delirium only represents a priority concern for 3% of critical care nurses.

As in previous research [54,55], they do not trust delirium detection tools, they do not feel comfortable with their use, they consider them arbitrary, and they report that their application has not been agreed upon with them, that it has been imposed. That is why they do not feel it as a shared measure, but as an obligation.

Despite the reluctance shown by the nurses interviewed in relation to delirium and the use of detection scales, our interviews point out the importance they attach to the management of environmental and/or family factors as preventive measures, as indicated by the evidence [56,57,58,59]. However, this care is not provided as specific and intentional care against delirium, but rather as measures of comfort or well-being for the child.

Several studies [60,61], indicate a high overall sensitivity of 94.1% (95% CI, 83.8–98.8%) and a specificity of 79.2% (95% CI, 73.5–84.9%), with a Cronbach’s α of 0.90, with a range of 0.87–0.90 for each of the eight items, indicating good internal consistency, a positive predictive value of 90% (95% CI, 79–97%), and a negative predictive value of 91% (95% CI, 80–97%). However, for inter-observer validity in the nursing population [62], the kappa coefficient between nurses was 0.60 (95% CI 0.44–0.76), indicating low to moderate agreement. The inter-observer reliability is especially low in the youngest infants (under 9 weeks), where the indices decrease to an ICC of 0.59 (95% CI 0.44–0.71), or in mechanically ventilated infants, where the ICC was 0.5 (95% CI 0.34–0.65). The Fleiss kappa for all infants was 0.47 (95% CI 0.34–0.6), which is a mild to fair agreement [63]. A study examining the validity of the CAP-D in children older than 6 months found mild to moderate agreement between the bedside nurse and the reference standard, with a kappa of 0.4 (95% CI 0.26–0.54) and a higher likelihood of discordant results when patients were younger (odds ratio [OR] = 1.1, 95% CI 1–1.2) and received some form of sedation ([OR = 4.1, 95% CI 1.5–11.5) [64].

Ultimately, these studies confirmed the reluctance shown by the interviewed personnel towards the use of the selected tool.

However, we must remember that reliability is not a fixed property of a measurement tool, but a product of the interactions between the tool, the subjects and the context of the evaluation, with the training of the observers being of special relevance in the use of the tool and its different variations as a strategy to increase the inter-observer reliability of the tool. In our case, this was not properly attended to, and the tool was selected without the “conviction” of those who should apply it, and without specific training on how to use it. All of this resulted into variations in observations, distrust in the tool, and disaffection with the detection program.

The problem regarding the knowledge and attitudes of health personnel, especially nurses towards the phenomenon of delirium, is common, and has been shown by some research, both in children [65,66,67] and adults [68,69], and intervention programs to have provided positive responses and changes in attitude and knowledge in the professionals involved [70,71].

As we find ourselves in a complex socio-work environment, the hypothesis of “resistance to change” in the nursing staff of the unit cannot be ruled out. While resistance has been considered a major barrier to the implementation of successful practice change in the popular literature, specific evidence on how it is a barrier within healthcare organizations is lacking [72]. However, resistance is a normal response to a threat to the baseline status quo. Nurse leaders prepared with knowledge of resistance, including background and attributes, can minimize the potential negative consequences of resistance and capitalize on the powerful impact of adapting to change.

This phenomenon is not new when dealing with changes in the nursing work structure. It has been estimated that more than half of all organizational change projects are unsuccessful [73].

Even in the presence of evidence in favor and the goodness of new care practices, some nurses may be reluctant to implement them or to have positive attitudes toward them.

Among the justifications for this phenomenon, we find rapidly changing regulatory requirements, technological innovation, and a constant flow of new information and knowledge that may present nurses with challenges they are reluctant to embrace.

Individuals fear the unknown and even experience a kind of loss associated with change, an increase in personal demands that they are unwilling to take on. Mistrust of leaders, lack of communication, or lack of clarity in group goals and objectives fuel the phenomenon, further increasing resistance to change [72].

Among the justifications that are provided in the evidence to show resistance to change, we find traditional ways of working that work well (“we have always done it this way without problems… I was trained this way…this is how we do things here…”) lack of confidence in the evidence shown in favor of change (“I do not trust the evidence…. We do not believe it makes a difference…”), and lack of time (“we cannot make so many changes at once… We are in a hurry… we are short of time…”) [73].

For changes to be accepted, there needs to be a “change agent”. When nurses are part of the decision-making process, organizations are much more likely to turn “enemies” into allies, and, at the same time, decrease the general anxiety around change [74].

More effective strategies include the application of the techniques or assistance protocols in a tutored manner at the outset, adequate and prompt training programs, visual guides or tutorials with the new procedures, etc. In addition, management and leadership positions can implement “pep-talk” groups or meetings, where staff can talk about the evidence for change and discuss the benefits or drawbacks of new practices.

Therefore, we can sense that while resistance to change is a common problem for care teams facing a new task that is seen as a “burden”, it is the responsibility of management and leadership teams to manage the incorporation of new practices with safe strategies to avoid this phenomenon. In short, motivating and training staff is one of the keys to success.

Regarding the apparent confusion shown by the interviewees between delirium and withdrawal syndrome observed in children, due to the use of opioid drugs (Iatrogenic), this is not strange, but rather a frequent one among clinicians. Both of them are not mutually exclusive, and can coexist in the same patient at the same time, especially for hyperactive delirium [75]. The rating scales of both diagnostic labels have common aspects with considerable overlap [76]. This may be due to a deficient degree of knowledge regarding these different diagnostic labels, and their defining causes and forms of identification [77].

As limitations, we recognize that this study was conducted in a single PICU of a public hospital in the southeast of Spain. Therefore, the findings of this study may not be the same as those found in other hospitals around the country. That is why studies based on mixed-type methodologies, with the inclusion of a representative sample from other centers, are necessary to reaffirm the findings, or to consider them as a local phenomenon.

For reasons of time pressure, the transcribed testimonies were not given back to the interviewees, so that they could corroborate their testimonies as true or modify what was reflected in the transcripts them in some manner. This is a practice that contributes to the process of transparency and truthfulness of qualitative research, and should be implemented as a good practice measure. In our case, it was not performed. However, the impartiality of the researcher transcribing and analyzing the interview data in the first phase was maintained throughout the process.

Finally, we understand that the non-use of specific qualitative analysis software limited the possibility of specific analysis of interview text, such as the “word cloud”. This tool helps us, in a visual and clear way, to understand what the subject feels about a topic/situation, as it allows them to summarize their view of a topic, and key words emerge on the surface.

## 5. Conclusions

In the hospital unit studied, we found negative attitudes, values, and beliefs of the PICU nursing staff towards infant delirium, especially towards the use of the measurement instruments used (DPCA), and the manner in which the action program was implemented. These negative attitudes partly explain the failure of the delirium detection program implemented in the unit.

It has been shown that there is a detachment of the PICU nursing staff towards the phenomenon of delirium, among other things, due to the lack of training in this regard, conceptual confusion (withdrawal syndrome with hypoactive delirium), and interprofessional conflicts (doctors-nurses).

The actions carried out in the delirium detection program were seen by the nursing professionals of the unit as measures imposed by the medical community, not thinking of them as a priority of care for the sick child. Early detection of delirium is not assumed to be a priority in their work as nurses.

These results are consistent with diverse evidence that indicates that the success of an intervention program depends on the choice of the correct tool, the adequate training of professionals, teamwork, and the joint mission, all carried out prior to the implementation of corrective measures. This did not occur in the investigated context.

Thus, the challenges when implementing a delirium detection program start from aspects such as staff motivation, sharing common therapeutic objectives among all professionals involved, regardless of their category, specific training, and continuous support. All of this assumes the ultimate responsibility of both the nurses responsible for applying the measures and the program managers.

## Figures and Tables

**Table 1 healthcare-12-00052-t001:** Characteristic of the sample.

ID	Age (Years)	Nursing Experience (Years)	PICU Experience (Years)	Adult ICU Experience (Years)	Specific Training Pediatric Nurse
N1	26	4	1	No	Yes
N2	25	4	1	No	Yes
N3	28	8	3	No	Yes
N4	35	13	6	No	Yes
N5	37	13	7	No	Yes
N6	35	14	15	Yes	No
N7	42	23	15	Yes	No

**Table 2 healthcare-12-00052-t002:** Main Codes and Themesc.

	First Order Codes	Code Definition	Main Themes
1	Specific Training Delirium for Nursing	Expressions related to nurses’ training on the subject of delirium are included here.	Training of professionals
2	Specific Training Delirium for Medicine	The expressions of the nursing professionals regarding the specific training of doctors on the subject of delirium are included here.	
4	Diagnosis of Delirium	The different definitions of delirium according to the opinion of the professionals are included here.	Childhood Delirium in ICU.
5	Definition of Delirium	The different comments defining delirium and describing its characteristics according to the participants are included in this code.	
6	Pharmacological Treatment of Delirium	Comments referring to the different drugs used to treat delirium are included in this code.	
7	Non-pharmacological Treatment of Delirium	This category includes activities and measures used for the non-pharmacological treatment of patients with delirium.	
8	Delirium Incidence	It collects the incidence data according to sex and age of delirium in the unit’s patients.	
9	Delirium Prevention Measures	Collects the activities carried out by the nursing team for the prevention of delirium in patients.	
10	Delirium Registry	Collects the additional resources used to record delirium, the characteristics of delirium in each patient, as well as extra data that could be relevant.	
11	Treatment Withdrawal Syndrome	Pharmacological measures used for treatment.	Abstinence Syndrome in ICU.
12	Withdrawal Syndrome Symptoms	This category collects the symptoms of withdrawal syndrome that can be observed in patients.	
3	Confusion between Withdrawal Syndrome and Delirium	Nurses’ expressions and ideas about the confusion between Withdrawal Syndrome and Delirium.	
13	Limitations Scales	They describe the drawbacks and limitations of the scales used to measure delirium.	Use of Diagnostic Tools.
14	Criteria for Passing Scales	The indications and situations in which these scales are used, as well as the nursing staff’s own criteria, are included.	
15	Validity Scales	The opinions of nursing professionals on the validity and reliability of the scales measuring delirium are included.	

## Data Availability

The data are available within the article.

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
