# Peer review of "Challenges of the Implementation of a Delirium Rate Scale in a Pediatric Intensive Care Unit: A Qualitative Approach"

_healthcare, 2023, doi:10.3390/healthcare12010052_

Round 1

Reviewer 1 Report

Comments and Suggestions for Authors

The study "Challenges of the Implementation of a delirium rate scale in a Pediatric Intensive Care Unit. A Qualitative Approach" is about challenges faced by the healthcare staff. There are some improvements needed before going for the decision.

1. Lines 29-38 are suggestions for writing a good introduction but the author did not relate their research with it. If so, it can be removed.

2. Similarly, lines 207-211 are also about the general importance of the interviews. It may be removed.

3. The time of interviews was during the COVID-19. Did the authors observe special measures/questions/precautions to prepare the interviews? How this study is related to post-covid time is unclear.

Author Response

Thank you very much for your contributions.

In the attached text you will find the responses we have given to your comments.

I hope you like them.

Thanks for your comments

Reviewer 2 Report

Comments and Suggestions for Authors

An attractive manuscript has been submitted. Good results have also been extracted. I reviewed the manuscript. Introduction seems long. Why is this research done qualitatively? Couldn't the authors investigate the reason for not using tools related to delirium in children's departments with quantitative methods? How was the data analysis method? Is software used or is the analysis done manually? Was there a finding that could not be interpreted using the results of other findings? In other words, how has this research been able to expand the scope of existing science regarding the use of tools in special departments for children?

Author Response

(The authors gave the same response as above.)

Reviewer 3 Report

Comments and Suggestions for Authors

I read with the interest the manuscript.

General comments:

The manuscript presents the qualitative study conducted through semi-structured interviews with PICU Key Informants.  The aim of the study was to explore how nursing staff experienced the implementation of new program against delirium in one paediatric intensive care unit. It is a real- life clinical study exploring delirium which is one very important problem in all intensive care units and in paediatric population because we need more studies in children. Its additional strength is that the problem is studied from the perspective of nurses who experience and deal with this problem even more than doctors.

The  manuscript is relevant for the field, but presented in a rather unclear and not really well-structured manner.

Despite the very long Introduction, it is not clear what the concrete implemented program against delirium included.  I would suggest the authors to make the Introduction shorter but more detailed on the program elements  they implemented.  It is not clear what was the new in their clinical practice e.g. Use of scale(s), nurse training. The authors did not present any hypothesis. The aim could be more precise. Unfortunately, the authors did not make the study before the implementation, so there is no insight how was before the implementation of the program and this is the big weakness of the study.

The methodology is well described but characteristics of the study population were written through three chapters ( Introduction, Methods and Results). It would be easier to read the manuscript if the the authors would rearrange the manuscript and move the description of characteristics of the studied population written in the Introduction to the Methods. In addition I would suggest to incorporate the Table 1 and the Table 2  in the Methods.  

The manuscript’s results are reproducible based on the details given in the methods section.

The Conclusion is only partially consistent with the evidence and arguments presented.

The references are relevant.

The ethics statements and data availability statements are adequate.

Specific comments:

Line 2. Title should be in accordance with the name of the implemented program… “Challenges of the Implementation of a delirium rate scale….” Is it implementation of the scale, scales or something else”

Line 29-37: not belong to the manuscript

Line 319 . Table 1: the term “nickname” is not appropriate

Line 630-639: should be deleted; not consistent with the evidence and arguments presented in the manuscript.

Appendix 3:  written in Spanish; should be in English

Comments on the Quality of English Language

 Moderate editing of English language required .

Author Response

(The authors gave the same response as above.)

Reviewer 4 Report

Comments and Suggestions for Authors

Abstract:

Page 1, Line 8: Clarify the meaning of "His Unit."

Page 1, Line 14: Correct the spelling of "interviews."

Introduction:

Page 1, Lines 29-37: Explain the reason for including this paragraph in the introduction.

Page 1, Lines 41-43: Provide a description of delirium based on current classificatory systems such as ICD or DSM for better comprehension.

Page 2, Line 51-53: Ensure proper citation of references to support the statements made in the introduction.

Page 2, Line 68: Define "high metric properties" in the context of accuracy being 98.4%.

Page 2, Line 71: Explain the meaning of CAM-ICU.

Page 2, Lines 77-80: Rephrase the paragraph for better clarity.

Page 3, Line 119: explain the meaning of  "anti-delusion guidelines."

Page 3, Lines 126-135: Consider moving this content from the introduction to the methodology section under the subheading "setting."

The introductory section requires a comprehensive revision to rectify repetition, enhance coherence, and incorporate recent studies pertaining to the discernment of attitudes, values, and beliefs among nursing staff in the Pediatric Intensive Care Unit (PICU).

Methodology:

Page 173-174: Clarify the rationale behind using the snowball sampling method.

Page 5, Line 244: Specify if the COVID-19 pandemic had any impact on the study conducted in June 2021.

Page 7, Line 350:  explain the meaning of  "process of de-habituation to morphine-derived drugs."

General Comments:

1. Specify the individual or team accountable for employing diagnostic tools to identify delirium within this particular setting.

2. Augment comprehension by presenting data on delirium occurrences in the ICU throughout the past year.

3. Provide a rationale for abstaining from the utilization of statistical methods in streamlining data variables.

4. Define the precise scope of the study, whether it centers on delirium itself or the  delirium diagnostic tools.

5. Refine the discussion by concentrating on result interpretation, anchoring findings in pertinent previous research, and integrating relevant data and studies.

End

Comments on the Quality of English Language

Abstract:

Page 1, Line 8: Clarify the meaning of "His Unit."

Page 1, Line 14: Correct the spelling of "interviews."

Introduction:

Page 1, Lines 29-37: Explain the reason for including this paragraph in the introduction.

Page 1, Lines 41-43: Provide a description of delirium based on current classificatory systems such as ICD or DSM for better comprehension.

Page 2, Line 51-53: Ensure proper citation of references to support the statements made in the introduction.

Page 2, Line 68: Define "high metric properties" in the context of accuracy being 98.4%.

Page 2, Line 71: Explain the meaning of CAM-ICU.

Page 2, Lines 77-80: Rephrase the paragraph for better clarity.

Page 3, Line 119: explain the meaning of  "anti-delusion guidelines."

Page 3, Lines 126-135: Consider moving this content from the introduction to the methodology section under the subheading "setting."

The introductory section requires a comprehensive revision to rectify repetition, enhance coherence, and incorporate recent studies pertaining to the discernment of attitudes, values, and beliefs among nursing staff in the Pediatric Intensive Care Unit (PICU).

Methodology:

Page 173-174: Clarify the rationale behind using the snowball sampling method.

Page 5, Line 244: Specify if the COVID-19 pandemic had any impact on the study conducted in June 2021.

Page 7, Line 350:  explain the meaning of  "process of de-habituation to morphine-derived drugs."

General Comments:

1.Specify the individual or team accountable for employing diagnostic tools to identify delirium within this particular setting.

2.Augment comprehension by presenting data on delirium occurrences in the ICU throughout the past year.

3.Provide a rationale for abstaining from the utilization of statistical methods in streamlining data variables.

4.Define the precise scope of the study, whether it centers on delirium itself or the  delirium diagnostic tools.

5.Refine the discussion by concentrating on result interpretation, anchoring findings in pertinent previous research, and integrating relevant data and studies.

End

Author Response

(The authors gave the same response as above.)

Round 2

Reviewer 4 Report

Comments and Suggestions for Authors

may be accepted